# Investigating the impact of missing value handling on Boosted trees and Deep learning for Tabular data: A Claim Reserving case study

**Alexander Larionov**                                           *alexander.larionov15@imperial.ac.uk*
*Department of Mathematics*
*Imperial College London*

**Niall Adams**                                                          *n.adams@imperial.ac.uk*
*Department of Mathematics*
*Imperial College London*

**Kevin N. Webster**                                              *kevin.webster@imperial.ac.uk*
*Department of Mathematics*
*Imperial College London*

**Reviewed on OpenReview:** *https://openreview.net/forum?id=aV6dCg1VFV*

## Abstract

While deep learning (DL) performance is exceptional for many applications, there is no consensus on whether DL or gradient boosted decision trees (GBDTs) are superior for tabular data. We compare TabNet (a DL model for tabular data), two simple neural networks inspired by ResNet (a DL model) and Catboost (a GBDT model) on a large UK insurer dataset for the task of claim reserving. This dataset is of particular interest for its large amount of informative missing values which are not missing completely at random, highlighting the impact of missing value handling on accuracy. Under certain missing value schemes a carefully optimised simple neural network performed comparably to Catboost with default settings. However, using less-than-minimum imputation, Catboost with default settings substantially outperformed carefully optimised DL models - achieving the best overall accuracy. We conclude that handling missing values is an important, yet often overlooked, step when comparing DL to GBDT algorithms for tabular data.

## 1 Introduction

Many machine learning problems involve regressing or classifying with *tabular* or *structured* data. Since their introduction, GBDTs have performed well on such tabular data (Friedman, 2001). Meanwhile, DL has become the state of the art in many problems that involve *unstructured* data, e.g. images (He et al., 2016; Simonyan & Zisserman, 2015; Tao et al., 2020), audio (Ao et al., 2021), text (Baktha & Tripathy, 2017; Ziegler et al., 2019; Touvron et al., 2023), and their combinations (Radford et al., 2021; Rombach et al., 2022; Ramesh et al., 2021).

Naturally, with the rise of research in DL, architectures have been proposed that claim to outperform GBDTs on tabular data (Somepalli et al., 2021; Shavitt & Segal, 2018; Huang et al., 2020; Kadra et al., 2021). However, there is a lack of consensus on whether these architectures really are more accurate. Large studies (Grinsztajn et al., 2022; McElfresh et al., 2024; Borisov et al., 2022) have compared DL to GBDTs across many datasets, many tasks and different computational budgets and suggest that GBDTs are on average the more accurate model for tabular data. The proposed reasons for the performance edge of GBDTs in these large studies remains an area of active research. Theories include the ability of GBDTs to ignore irrelevant variables and model discontinuous functions (Grinsztajn et al., 2022). Importantly, neither the proposed DL

architectures nor the large studies investigate the impact of missing values. As missing values are common in real tabular data (Van Ness et al., 2023), and missing value handling can significantly impact the results of analyses (Jin et al., 2021), this is a significant gap in the literature.

In the remainder of this paper we use a claim reserving application with data from a large UK car insurer to shed light on the comparison of GBDTs to DL, with a particular focus on the impact of missing value handling. We specifically study this real-world dataset because it is characterized by extensive missing values that are explicitly *not* missing completely at random (MCAR) – offering a more realistic scenario than injection of MCAR missing values into previously studied complete datasets.

We investigate Catboost (Prokhorenkova et al., 2018) and three DL architectures: TabNet (Arik & Pfister, 2021) and two ResNet-inspired multi layer perceptrons (MLPs). TabNet was chosen as a specialised tabular DL architecture with prior validation in insurance (McDonnell et al., 2023). Catboost and ResNets MLP were chosen as the respective best GBDT and DL model from McElfresh et al. (2024).

In Section 2 we present background on our application as well as describe the car insurance dataset that we analyse. In Section 3 we give a more detailed description of the modelling strategies we compare. In Section 4 we cover three elements of methodology. Section 4.1 covers our approach to hyperparameter tuning, where extra care was taken to avoid bias. Section 4.2 covers the experiments used to investigate the impact of missing value handling. Section 4.3 briefly covers experiments studying injection of MCAR missing values into other previously studied complete datasets. In Section 5 we present and discuss the results on our insurance dataset, not only finding Catboost is the superior model for our data, but also highlighting the importance of missing value handling in model accuracy.

## 2   Background

Car insurance is an important financial service with a 2024 global value of over 1.9 trillion USD, which is estimated to reach over 2 trillion USD by 2028 (Statista, 2024). Car insurance works on the principle that insurers charge customers a *premium* in return for obligations to provide financial support in the event of contractually agreed risks. Accurate pricing is vital for both the sustainable profit of the insurer and fair prices for customers. The process of determining a price for a prospective customer in car insurance is complex (Olivieri & Pitacco, 2015; Werner & Modlin, 2010). It comprises of three core steps i) estimating the expected value of payments to the customer over the duration of the contract ii) estimating current liabilities for claims that are *reported but not settled (RBNS)* and iii) somehow sensibly combining the two prior estimates into a price. The first step typically comprises finding a model for *claim frequency* and a model for *claim severity*; the latter estimating the cost of a claim conditional on an accident. The second step comprises modelling the cost of claims conditional on them having already occurred and is called *claim reserving*. The third step combines the claim frequency, claim severity and claim reserve estimates using risk models and business considerations: such as profit margins, legal requirements, risk appetite and operational costs.

The focus within this work will be on the second step: claim reserving. Specifically, we focus on *outstanding claim reserve* modelling which is the process of predicting costs for claims that have been RBNS.

Typically, outstanding claim reserve modelling is mainly done on a portfolio level. In other words, insurance companies predict the overall reserve requirement for a given time period, say a quarter, across all customers. Importantly, these forms of claim reserve modelling use no individual claim information, instead using historic portfolio level settlement aggregates. This is done with deterministic algorithms such as run-off triangles, the chain ladder (CL) method and the Bornhuetter-Ferguson algorithm (Bornhuetter & Ferguson, 1972); or stochastic extensions of said algorithms.

We focus instead on *individual claim reserve modelling*, or *micro-level reserving*, an alternative method of reserving. Individual claim reserve modelling predicts portfolio reserves from aggregating estimates per incident. There is not yet a consensus that individual claim reserving is more or less accurate than aggregate modelling. Still, the hypothesised benefits of micro-level reserving are: greater insight into exposure profiles within a portfolio; more signal (i.e. relevant covariates) should produce more accurate models; and the

ability to adapt to trends that can be captured by covariates (Blier-Wong et al., 2021; Lopez et al., 2019; Delong & Wüthrich, 2020).

There is literature investigating the use of older machine learning (ML) algorithms such as CART (Breiman et al., 1984) and generalized linear models for individual claim reserving (Lopez et al., 2019; De Felice & Moriconi, 2019; Taylor et al., 2008; Wuthrich, 2018). Newer ML methods, such as neural networks (Delong & Wüthrich, 2020; Delong et al., 2022; Kuo, 2020) and GBDTs (Duval & Pigeon, 2019) have also had some, limited, research. These works analysing micro-level reserving strategies broadly conclude that their respective models are either on par or better than an aggregate CL method, validating micro-level reserving in principle. However, there are only a few such works; their insurance fields vary; they use small sets of covariates and some use simulated data. This makes it difficult to know whether the results are relevant to car insurance micro-level reserving. Furthermore, of considerable practical importance is that missing data is endemic to real insurance data (Fauzan & Murfi, 2018; Hanafy & Ming, 2021) and none of these works give any special focus to missing data. Finally, these works often report benchmarks against the CL method instead of overall accuracy which complicates the interpretation of results as disagreement with CL could be the consequence of more accurate modelling. To our knowledge, also noted by the survey of Blier-Wong et al. (2021), none compare modern ML methods directly to each other on real data. The lack of direct comparison means no conclusion can be drawn about the relative performance of newer ML methods for claim reserving.

Ultimately, both research in insurance and ML more broadly paints a blurry picture on the relative merits of GBDTs and DL for micro-level reserving using tabular data. Furthermore, treatment or influence of missing values on accuracy is not investigated when comparing the methods. Although missing value handling has been shown to be important in other fields (Herring et al., 2004) and as such could be important to reserving. This leaves reserving actuaries dealing with tabular data unclear on whether it is worth the investment to investigate and deploy these more modern ML algorithms nor the impact of missing data for said algorithms.

The most relevant work to ours, comparing DL to GBDTs in insurance, is McDonnell et al. (2023). They compare the DL architecture TabNet (Arik & Pfister, 2021) to the GBDT implementation XGBoost. They model discretised *claim severity* classification on a dataset with hundreds of thousands of claims. They find TabNet to be comparable to XGBoost, with marginally better F1 score. Although this is modelling claim *severity*, not claim *reserving*, we note that claim severity and claim reserving both model costs conditional on an accident occurring. However, claim severity is estimated *before* the accident occurs and claim reserve *after*. From the perspective of regression, the only difference between micro-level reserving and claim severity modelling is the number of covariates - as more is known after and accident occurs. In the work of McDonnell et al. (2023), although TabNet performs comparatively well to XGBoost, the models were evaluated on synthetic data generated using a neural network (So et al., 2021) thus potentially biasing performance towards DL as the model class was more likely to be correct. Furthermore, the casting of the regression problem into discretised classification and lack of missing data makes the findings less interpretable and transferable.

## 2.1 Data description

The tabular data we model in Section 4 consists of many hundreds of thousands of insurance claim feature vectors as rows, with hundreds of features as columns. This dataset has never been previously studied. The data is a combination of information available at policy issue (e.g. make and model of the car) and information available just after the time of claim reporting (e.g. accident date). The settlement value (SV) variable gives how much the insurer paid overall to settle a claim; inclusive of vehicle, personal and property damage. We aim to accurately predict SV for each claim to build a micro-level reserve, as described in Section 2. As we use supervised ML methods we only consider closed claims, i.e. there exists a SV to be used as a label.

Commercial confidentiality prevents us from giving a more detailed description of the data. However, we present the missing data properties in the next section. We present other data characteristics and their implications for modelling and data processing in Appendix A.1; this includes time varying properties and handling of high cardinality categoricals, such as postcode information.

### 2.1.1 Missing data

The dataset we study has extensive missing values. Over 50% of features contain missing values, therefore ignoring all features with missing values would drop the number of features by over half. This could drop highly informative features, e.g. details of additional drivers on a policy, which are missing in the majority of claims.

Furthermore, due to the interaction of missing values in multiple features there is *no complete feature vectors*, i.e. every row has at least one missing value. Therefore if we directly apply a strategy such as complete-case analysis (Little & Rubin, 2019, p. 47), where any row with missing values is dropped, the whole dataset would be dropped. Instead, we can first drop features that are missing in more than a certain proportion of cases and then run a complete-case analysis. This latter approach is also used to deal with missing values by Grinsztajn et al. (2022), one of the broad comparative studies mentioned in Section 1. We explore this method, along with alternative imputation approaches, calling this missing value handling strategy `Drop` in Section 4.

Beyond the extent of missing values, the data presents a dependence of the response, SV, on the missing value structure. This can be shown by a large shift in the mean and standard deviation of the SV when using `Drop` at various missing value proportion thresholds. Smaller proportions of missing values in a feature vector are associated with substantially higher SV. This suggests that the data is not MCAR (Little & Rubin, 2019, p. 13-23). Therefore, fitting a model under a `Drop` strategy will result in biased predictions, above and beyond any bias introduced by the model or training algorithm. This bias also means the accuracy results of a model fit on `Drop` are not comparable to those of a model fit on imputed data.

To summarise, missing values represent a large portion of our insurance dataset and are *not* MCAR as the SV varies substantially conditional on the missing value structure. This highlights the importance of investigating and choosing appropriate missing value handling strategies.

## 3 Models

In this section we start by defining notation, outlining some DL terms and then briefly give background on the models used: i) Catboost, a GBDT model; ii) two ResNets, a general purpose feed forward DL model: ours and that of Gorishniy et al. (2021); and iii) TabNet, a DL architecture specifically designed to accommodate tabular data. Within this section we do not aim to provide comprehensive details. Instead we aim to describe methods in sufficient detail to follow the hyperparameters tuned in Section 4.1.

### 3.1 Notation

We denote the dataset $\mathcal{D}$, as a set of tuples, $\mathcal{D} = \{(\mathbf{x}_k, y_k)\}_{k=1}^{N}$, where $y \in \mathbb{R}^{+}$ denotes the target settlement value, $N$ denotes the number of claims and $\mathbf{x}$ denotes a feature vector with $D$ features. Subscripts denote indexing on an arbitrary ordering of data tuples from the overall dataset.

We seek a model, $F(\mathbf{x})$, to predict the claim settlement value $y$. The accuracy of this model is measured by some loss function $L(y, F(\mathbf{x}))$, that we wish to minimise.

### 3.2 Catboost model

Catboost (Prokhorenkova et al., 2018) is a GBDT (Friedman, 2001) with a special procedure for categorical encoding and gradient estimation. Note that the Catboost algorithm details are complex and have many configurable options. Here we only cover the relevant details of the base GBDT algorithm and briefly mention the core novel concepts proposed by Prokhorenkova et al. (2018). For removal of ambiguity, as the default behaviour can vary depending on the execution hardware, we present details and use defaults for running on a CPU opposed to a GPU.

### 3.2.1 Gradient boosted decision tree

Boosted models learn an additive ensemble of 'weak learner' models. If $T$ is the total number of weak learners we want to use, the boosted model would be:

$$F_T(\mathbf{x}) = G_0(\mathbf{x}) + \sum_{t=1}^{T} \eta G_t(\mathbf{x}), \tag{1}$$

where $G_t(\mathbf{x})$ is the $t$th 'weak learner', $\eta$ is a weighting factor, and $G_0(\mathbf{x})$ is an initial estimate, such as the mean response of the training set.

Usually boosted models are built in a sequential fashion, e.g. the $i$th model incorporating $i$ weak learners would be $F_i = F_{i-1} + G_P$ for $i = 1, ..., T$. The sequential construction of the model enables the procedure to be terminated early if validation performance is not improving i.e. return $F_i(\mathbf{x})$ with $i < T$.

For gradient boosting, the summands $G_t(\mathbf{x})$, $t \in \{1, ..., T\}$, are chosen from within a hypothesis class of functions $\mathcal{G}$ to approximate $-\left.\frac{\partial L}{\partial F}(y, F(\mathbf{x}))\right|_{F_{t-1}}$, the negative functional derivative of the loss. This negative functional derivative of the loss is also called a pseudo-residual and denoted $r_{t-1}(y, \mathbf{x})$ (Friedman, 2001).

To evaluate $r_{t-1}(y, \mathbf{x})$ requires knowledge of both $\mathbf{x}$ and $y$. As $y$ is unavailable outside the training set, $r_{t-1}(y, \mathbf{x})$ can only be evaluated on the training data. However we can approximate $r_{t-1}$ with a given summand $G_t(\mathbf{x})$ and measure of function fit, $L'\big(r_{t-1}(y, \mathbf{x})$:

$$G_t = \underset{G \in \mathcal{G}}{\arg\min} \sum_{(\mathbf{x}, y) \in \mathcal{D}} L'\big(r_{t-1}(\mathbf{x}, y), G(\mathbf{x})\big). \tag{2}$$

In practice finding the true $\arg\min$ is infeasible so $G_t$ is some approximation learned following a standard algorithm to minimise $L'$.

$L'$ can be different from $L$ as it is used to fit $G_t(\mathbf{x})$ to $r_{t-1}(y, F(\mathbf{x}))$ to enable derivative evaluation on data outside the training set. It is the addition of $G_t$ to the ensemble that contributes to the minimisation of $L$ given a small enough step size $\eta$.

This results in the boosted ensemble approximating a gradient descent of the loss functional (in the space of linear combinations of $\mathcal{G}$) with constant learning rate $\eta$:

$$F_t = F_{t-1} + \eta G_t \approx F_{t-1} - \eta \left.\frac{\partial L}{\partial F}\right|_{F_{t-1}} \tag{3}$$

In the context of GBDTs; the weak learner is a decision tree (Breiman et al., 1984). The choice of step size $\eta$ and desired ensemble size $T$ are among the hyperparameters tuned in Section 4.1.

### 3.2.2 Catboost: Pseudo-residual calculation and categorical encoding

Catboost aims to improve performance on unseen data by reducing overfitting. The key innovations of Catboost are twofold: i) how the pseudo-residuals, $r_{t-1}$, are approximated using $G_t$ and ii) how categorical variables are encoded. Although we will describe the core idea of the improvement, there are further technicalities and engineering modifications present in Prokhorenkova et al. (2018), e.g. to improve speed, that we do not describe.

For the alteration to $G_t$ fitting, the core idea is to fit $G_t$ on data excluding the data point for which it will predict $r_{t-1}$, i.e. to calculate $G_t(\mathbf{x}_k)$ Catboost would fit $G_t$ on data $\{\mathbf{x}_j : j < k\}$. This excludes the data point $\mathbf{x}_k$, and also generates different $G_t$ for different data points.

Likewise, Catboost follows this procedure for generating a categorical encoding. For a given data point $\mathbf{x}_k$, Catboost fits a target mean encoding (Pargent et al., 2022) on a discretized target for $\{\mathbf{x}_j : j < k\}$ that are before the point encoded. Furthermore, when processing categoricals Catboost uses a novel algorithm to redefine category labels as the algorithm runs ('feature combinations' in the original work).

We note that there are further important implementation details regarding the categorical encoding, such as how the target is discretised prior to mean encoding, that are absent from the original publication. Full details can be found in the tool's documentation (Catboost, 2024b) and codebase (Catboost, 2024a).

This work studies Catboost with two different categorical encoding schemes: first with target mean encoding, and second with Catboost's novel debiased target encoding, that also employs category redefinition.

### 3.3 ResNet MLP

A ResNet, short for residual network, MLP is a feed-forward neural network (Murphy, 2022, p. 419) with additive *residual* connections that skip layers. Without additional knowledge about the underlying structure of the data, an MLP is a simple general purpose DL architecture; and skip connections make training more stable (Murphy, 2022, p. 445).

We implement a ResNet by using residual connections across building blocks, along with skip connections to the output. We use a building block layout of `BatchNorm`, `ReLU` and `Dense` as in He et al. (2016). We add `Dropout` following the example of Gorishniy et al. (2021) – which proposed the best performing DL model (a ResNet) from McElfresh et al. (2024). For the sake of clarity, our ResNet MLP is *not* identical in architecture to Gorishniy et al. (2021), and by extension McElfresh et al. (2024). For details of the differences with Gorishniy et al. (2021) see Appendix A.3. To contextualise our findings we also present results using the ResNet architecture from Gorishniy et al. (2021), referring to it as 'RTDL ResNet'.

Figure 1 shows the layout of our ResNet MLP layers on the far left, with their combination into a sub-block denoted by $A$. The ResNet sub-block, $A$, is repeated in a residual pattern to form a high level block $B$ – shown on the right of Figure 1. This higher level block $B$ is in turn composed using skip connections into an overall model. Each block has independently trainable parameters. In the context of deep learning, choosing the architecture size (such as number of blocks, size of `Dense` layers in units etc.) is a part of the broader problem of hyperparameter tuning. Our approach involves choosing the number of $B$ blocks to vary depth; and choosing the number of units used in every `Dense` layer to vary the width of the network. This tuning is further described in Section 4.1. We use a grid search, described in Section 4.1.3, to select the optimiser used, the initial learning rate of the optimiser, learning rate schedule (Murphy, 2022, p. 288), `Dense` layer weight regularisation strategy and regularisation intensity.

### 3.4 TabNet model

TabNet (Arik & Pfister, 2021) is a DL architecture specifically designed for tabular data. TabNet works by learning a *step* that multiplies a subset of features by zero, conditional on the input. Then TabNet uses DL layers on the remaining non-zero parts to produce an intermediate *decision* vector. The architecture sequentially applies multiple steps. Each step can determine a different subset of features to set to zero – so a feature that is set to zero for one step does not need to be zero for the following steps. In fact, the hyperparameters described below control how many distinct features can be selected and their potential for reuse across steps. As each step can select different subsets, each step can produce a different decision vector. Finally, the decisions from all steps are combined through a DL layer into a prediction.

TabNet is a complex architecture for which we defer the detailed description to the original paper (Arik & Pfister, 2021). However, there are some key hyperparameters which we are required to tune.

The number of steps, $S$, determines the number of different feature subsets that are modelled to produce a prediction. With $S = 1$ only a single subset of the features is used, with more steps resulting in more feature subsets. Intuitively, more steps increases the overall number of features used, but also increases the depth of the network and destabilises training.

The so-called 'relaxation parameter', $\gamma \geq 1$, is designed to encourage different feature subsets to be selected at each step. When $\gamma = 1$ TabNet has the special property of being able to prevent reuse of features between steps. As $\gamma$ increases, TabNet is more able to reuse features between steps.

The sparsity regularisation coefficient, $\lambda \geq 0$, is used to encourage more input features to be zeroed out in each step. As $\lambda$ increases the network can zero out more features, even if it decreases training accuracy.

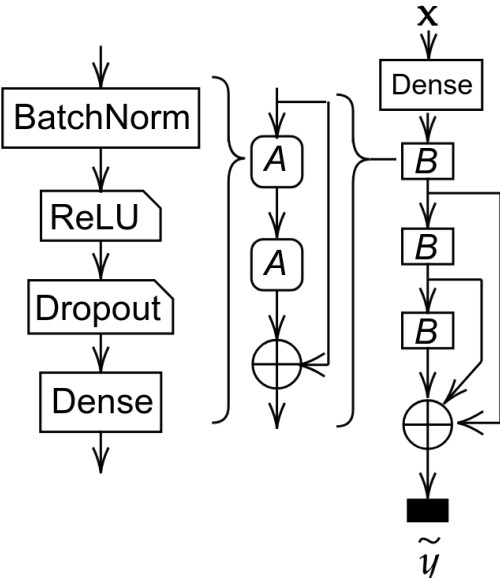

Figure 1: ResNet MLP model architecture. The architecture consists of a number of blocks, $B$. The skip connections in the diagram indicate that the outputs from each block $B$ are all summed together and passed through a final Dense layer to produce the scalar output $\tilde{y}$. Each layer $B$ consists of three sub-blocks denoted as $A$ in the diagram. Each $B$ contains a single residual connection so that the block output is produced by summing the outputs of the final two sub-blocks $A$. Each sub-block $A$ consists of feed forward `BatchNorm`, `ReLU`, `Dropout` and `Dense` layers.

## 4  Experimental method

To investigate the impact of preprocessing schemes for handling missing values, we first tuned the hyperparameters of each model. Preliminary analysis, described in Appendix A.5, suggested the best hyperparameters did not vary with preprocessing scheme. Therefore, the hyperparameter tuning process was done independently of later missing value investigation.

To prevent data leakage, the last 15% of the data was set aside into a test set, $\mathcal{D}^{\text{test}}$, shown in Figure 2. This $\mathcal{D}^{\text{test}}$ was always withheld from training or validation procedures and only used to report the metrics presented in Section 5.

Early stopping (Murphy, 2022, p. 448) was applied to improve training speed and prevent overfitting in both hyperparameter tuning and final model training. Early stopping is a form of regularisation where out-of-sample model performance is evaluated at regular intervals on a dataset withheld from training. When the performance on the withheld dataset decreases, the training algorithm is terminated. Preliminary analysis confirmed there was no decrease in accuracy from using early stopping.

Section 4.1 describes the tuning of hyperparameters discussed in Section 3. Section 4.2 describes the experimental setup used to investigate the impact of missing value handling and categorical encoding. Practical commentary on the training speeds of the algorithms can be found in Appendix A.2.

### 4.1  Hyperparameter tuning

Hyperparameter tuning on our ResNet and TabNet consisted of a grid search optimising for accuracy. For each modelling strategy a hyperparameter (HP) grid was subjectively chosen after initial trial and error. Although grid search may be a common practice in industry, it is also well known to theoretically underperform more principled methods of HPO such as Optuna (Akiba et al., 2019). As such, we also reran analyses for our ResNet using Optuna with the same upper and lower bounds on numeric hyperparameter

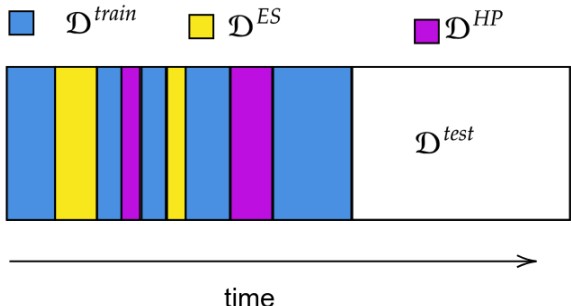

Figure 2: Dataset partitioning strategy for both hyperparameter tuning and final evaluation, where $\mathcal{D}^{\text{ES}}$ and $\mathcal{D}^{\text{train}}$ are shuffled per replication of a given experiment. $\mathcal{D}^{ES}$ is a split of data used for early stopping. $\mathcal{D}^{HP}$ is a split of data used for evaluation of hyperparameters. $\mathcal{D}^{HP}$ is sampled randomly in time, $\mathcal{D}^{test}$ is exclusively future data.

values. Those results can be found in Appendix A.4 but we note that in this instance Optuna hyperparameter optimisation (HPO) did not significantly change performance. For consistency with McElfresh et al. (2024), Optuna was used for HPO on the RTDL ResNet model and presented in Section 5, Table 1.

A single data subset, $\mathcal{D}^{\text{HP}}$, was sampled once for accuracy evaluation of *all* models under any given HP configuration. For clarity, $\mathcal{D}^{\text{HP}}$ was *not* a future partition of the data, but rather randomly sampled in time.

For each node in the grid, we randomly split the remaining data (after removing $\mathcal{D}^{\text{test}}$ and $\mathcal{D}^{\text{HP}}$) into $\mathcal{D}^{\text{train}}$ and $\mathcal{D}^{\text{ES}}$ subsets, illustrated in Figure 2. Training was performed solely on $\mathcal{D}^{\text{train}}$ whilst $\mathcal{D}^{\text{ES}}$ was used to trigger early stopping (ES). Once training was completed, the root mean squared error (RMSE) for a given node was evaluated on $\mathcal{D}^{\text{HP}}$. This random splitting of $\mathcal{D}^{\text{train}}$ and $\mathcal{D}^{\text{ES}}$, and subsequent evaluation of RMSE on fixed $\mathcal{D}^{\text{HP}}$, was independently repeated 10 times for each HP node in the grid. This repeated split sampling is called Monte-Carlo cross-validation (Kuhn et al., 2013, p. 71-72).

The best HPs for a given model were chosen on the basis of the lowest RMSE averaged across the 10 Monte-Carlo cross-validation samples. This best HP configuration for a given model was then used for the model-to-model comparison as described in Section 4.2.

The use of separate data subsets for early stopping, $\mathcal{D}^{\text{ES}}$, and accuracy evaluation, $\mathcal{D}^{\text{HP}}$, allowed us to remove bias associated with evaluation on data that was indirectly used for training. This is an especially rigorous process in contrast to what is often done in practice, where one validation set would be used for both early stopping and evaluation. However, as we had sufficient data, we opted for separate subsets to minimise potential bias.

Further details of the training algorithm and hyperparameter selection for each modelling strategy follow.

### 4.1.1 Catboost hyperparameter tuning

Initial manual exploration of the hyperparameters gave no significant improvements in $\mathcal{D}^{\text{HP}}$ RMSE. The step size, $\eta$, was varied between 0.005 and 0.018. The ensemble size, $T$, was varied between 1000 and 10000. Mean squared error was used as the loss function, $L$. As no noticeable improvement came from heuristic tuning, a complete grid search was not performed and all hyperparameters were left as defaults for evaluation of Catboost in model-to-model comparison. The only non-default choice was inclusion of early stopping which was used to speed up training, and had no noticeable effect on accuracy in the hyperparameter tuning stage.

### 4.1.2 TabNet tuning

For TabNet we follow the original paper (Arik & Pfister, 2021) in using the Adam optimiser (Kingma & Ba, 2014) and an exponential decay learning rate schedule with a fixed initial learning rate of 0.001. The

sparsity regularisation coefficient, $\lambda$; number of steps, $S$; and relaxation parameter $\gamma$ were tuned. Following the recommendations and ablations of Arik & Pfister (2021): $\lambda$ was varied between 0.0001 and 0.01, $\gamma$ was varied between 1.3 and 2 and $S$ was varied between 3 and 10.

### 4.1.3 ResNet MLP tuning

Currently, there is no principled way to construct a deep learning architecture, for example choosing depth and width, beyond intuition; trial and error; and neural architecture search (Ren et al., 2021). As neural architecture search was prohibitively expensive from a computational perspective we instead opted for a heuristic trial and error approach for choosing our ResNet architecture.

We began by arbitrarily choosing some expressive high level block, denoted $B$ in Figure 1. We built this block from sub-blocks, denoted $A$ in Figure 1. Sub-blocks $A$ utilise a ResNet (He et al., 2016) layout of layers as described in Section 3.

We then chose the depth: i.e. the number of $B$ blocks composed together prior to a `Dense` layer with a single output unit, with no activation, for prediction. A depth of 3 was chosen; subjectively balancing simplicity of the model with expressive power.

We then parameterised the width of a block with $d$, the number of units in each `Dense` layer of the ResNet block, denoted $A$ in Figure 1. We varied $d$ between heuristically identified limits wherein the model exhibited underfitting and the capacity to overfit training data. Underfitting was identified by both the training and validation error being similar and approximately constant per training epoch. Capacity to overfit was identified by the ability for the model to keep reducing training error whilst validation error is constant or getting worse. These criteria were considered fulfilled when the validation loss was not improving whilst training loss was still decreasing after 1000 epochs.

Varying $d$ within the bounds of under and overfitting did not noticeably impact accuracy. As such, $d$ was chosen for a total model size of four hundred thousand parameters. This was between the number of parameters which under- or overfitted. After the architecture was heuristically selected, grid search hyperparameter optimisation followed. The optimisers compared were Adam (Kingma & Ba, 2014) and RMSProp (Tieleman & Hinton, 2012). Both L1 and L2 regularisation of kernel weights were used; coefficients for each were varied between 0.01 and 1. Exponential Decay (Murphy, 2022, p. 288) and Cosine Decay (Loshchilov & Hutter, 2016) learning rate schedules were both compared; where initial learning rate was varied between 0.001 and 0.01 for each.

## 4.2 Investigating different preprocessing schemes

We aimed to investigate the impact of missing value handling through the comparison of each model under different preprocessing schemes.

To enable a pairwise comparison of trained models, the data subsets for these experiments were different from those of the hyperparameter tuning in Section 4.1. Instead of sampling different subsets per node in a grid search *all* experiments used the same set of 20 Monte-Carlo cross-validation samples of $\mathcal{D}^{\text{train}}$ and $\mathcal{D}^{\text{ES}}$. These were sampled once prior to running any experiments, enabling pairwise model comparison. As evaluation was performed on $\mathcal{D}^{\text{test}}$, there was no need for another withheld evaluation partition, like $\mathcal{D}^{\text{HP}}$.

A given $\mathcal{D}^{\text{train}}$ subset was used for training all models with all missing value handling schemes, described below, and the corresponding $\mathcal{D}^{\text{ES}}$ was used to trigger early stopping. After the models were trained, RMSE was evaluated on $\mathcal{D}^{\text{test}}$ for each of the 20 $\mathcal{D}^{\text{train}}$ and $\mathcal{D}^{\text{ES}}$ samples. The mean and standard error of the RMSE across the 20 Monte-Carlo cross-validation samples for each model and preprocessing scheme are reported in Table 1. These results are discussed in Section 5.

### 4.2.1 Missing value handling

Four missing value handling strategies were investigated. Two of the methods, `Drop` and `Binarize`, were used in an attempt to disentangle the capability of a model to successfully extract signal from observed covariates and from missing value structure. The other two, `LT Min Impute` and `Mean Impute`, were used to

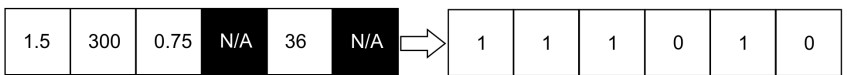

Figure 3: Example row undergoing `Binarize` transformation; where 'N/A' represents missing values. Present covariates map to 1; missing to 0.

investigate model capability to jointly use covariates and missing value structure through imputation. The missing value handling strategies are summarised:

- `Drop`: Rows with any missing values were dropped after first removing features with more than 30% missing values – as described in Section 2.1. From this scheme we aimed to isolate the ability of each modelling strategy to extract signal from observed covariates.

- `Binarize`: The whole dataset was converted into a binary representation of whether the covariate was observed or not, see Figure 3. This retained only the signal inherent in the missing value structure. With this scheme we isolate the ability of model strategies to regress against missing value structure.

- `LT Min Impute`: Missing values were imputed using less than the minimum of the numeric feature. This is the default missing imputation strategy of Catboost. The use of this strategy allows the weak learner trees, $G_t$ in Section 3, to separate the missing values completely from observed values. However, this also means that should the tree split on an observed value it will include all missing values on the lesser side of the split.

- `Mean Impute`: Missing values were imputed using the mean of the corresponding numeric feature. This was the approach used in McElfresh et al. (2024).

We also present results using *both* the `LT Min Impute` and `Binarize` strategies simultaneously in Appendix A.7 – finding there is no noticeable benefit to combining the approaches over `LT Min Impute` for our dataset.

### 4.2.2 Categorical encoding

Categorical encoding was performed using target mean encoding for the majority of the experiments, this compared the modelling strategies fairly by holding potential confounders in categorical encoding constant.

In the interests of identifying the best performance, we also evaluated the original Catboost categorical encoding with the Catboost GBDT algorithm, described in Section 3.2. We report the model names in Table 1 with the suffix(CE) indicating that Catboost encoding was used; otherwise mean encoding was used.

### 4.3 Effects of imputation scheme on other datasets

To study the broader relevance of results obtained from our insurance dataset, further analyses were performed on two other datasets under `Drop`, `Mean Impute` and `LT Min Impute`. The leading two GBDTs and leading two DL algorithms from McElfresh et al. (2024) were compared on two datasets using the TabZilla framework. The datasets did not contain missing values; so MCAR missing values were simulated by removal. As MCAR data are not the focus of this case study we present the results in Appendix A.6. Overall, these preliminary analyses showed lesser impact from missing value handling than our not MCAR application but still demonstrate the relevance of our work.

## 5 Results and discussion

The results for various preprocessing strategies are presented in Table 1. As a benchmark, we include a `Mean prediction` row which shows the performance that is obtained by using the mean SV of a given training dataset as a constant prediction, using no feature information. Note the performance evaluated with the

`Drop` preprocessing scheme is not directly comparable to the performance for other preprocessing schemes, i.e. `Drop` cannot be compared with other strategies across rows in Table 1. This is because the missingness mechanism is not MCAR and causes a distributional shift within both the training and evaluation data when dropping rows. However, due to consistency of the training datasets, these results are comparable within modelling strategies, i.e. one can compare down columns of Table 1.

Table 1: $\mathcal{D}^{\text{test}}$ RMSE mean and standard error, *se*, over cross validation partitions. **Lower RMSE is better**. RMSE is to the nearest integer, *se* is to 2 significant figures. (CE) indicates Catboost encoding was used for categorical encoding. † replicates the HPO, architecture and imputation scheme of McElfresh et al. (2024).‡ corresponds to an Optuna HPO scheme.

| Model | Drop | $(\pm se)$ | LT Min Impute | $(\pm se)$ | Mean Impute | $(\pm se)$ | Binarize | $(\pm se)$ |
|---|---|---|---|---|---|---|---|---|
| Catboost (CE) | 1969 | $(\pm 10)$ | 1452 | $(\pm 6.3)$ | 1519 | $(\pm 17)$ | 2302 | $(\pm 0.18)$ |
| Catboost | 2060 | $(\pm 1.7)$ | 1574 | $(\pm 2.1)$ | 1524 | $(\pm 5.2)$ | 2268 | $(\pm 0.28)$ |
| Our ResNet | 2046 | $(\pm 1.2)$ | 1872 | $(\pm 4.5)$ | 7855 | $(\pm 1600)$ | 2288 | $(\pm 1.8)$ |
| RTDL ResNet‡ | 2439 | $(\pm 24)$ | 2007 | $(\pm 20)$ | 2387 | $(\pm 270)$† | 2282 | $(\pm 2.4)$ |
| TabNet | 2702 | $(\pm 8.1)$ | 2748 | $(\pm 5.3)$ | 3081 | $(\pm 180)$ | 2754 | $(\pm 5.1)$ |
| Mean prediction | 2813 | $(\pm 0.34)$ | 2764 | $(\pm 0.020)$ | 2764 | $(\pm 0.020)$ | 2764 | $(\pm 0.020)$ |

The TabNet row of Table 1 shows that TabNet substantially underperforms both Catboost and the ResNet MLP. Notably, TabNet performs on par with the `Mean prediction` benchmark. This indicates that TabNet is either not suitable for micro-level reserving with this dataset or, at best, that TabNet is very difficult to train. This runs counter to published work on the use of TabNet for claim severity modelling (McDonnell et al., 2023), but in line with surveys of DL for structured data (Grinsztajn et al., 2022; McElfresh et al., 2024) where TabNet performed poorly. As mentioned in Section 1, this could potentially be explained by the fact that the results of McDonnell et al. (2023) could be biased because the modelled data is itself generated from a neural network.

The relative accuracy of the RTDL ResNet MLP row of Table 1 agrees with McElfresh et al. (2024) in outperforming TabNet and underperforming CatBoost. Although for `Drop`, `LT Min Impute` and `Binarize` the RTDL ResNet has lower accuracy than our ResNet it is interesting to note that under `Mean Impute` the RTDL ResNet performs substantially better than any other DL algorithm. This highlights the sensitivity of previously studied algorithms to imputation schemes.

However, we note that the performance of `Mean Impute` is still generally worse than that of `LT Min Impute` across most models. `Mean Impute` demonstrates moderately high standard error for the RTDL ResNet and high mean RMSE and standard error for our ResNet and TabNet. The exceptionally high RMSE for our ResNet appeared to be due to the rare prediction of low value claims many orders of magnitude larger than they were. It is unclear why this would be the case specifically for `Mean Impute` and our ResNet. We hypothesise this could be due to `Mean Impute` making it difficult to distinguish meaningfully missing data, where absence of data is not due to random lack of records but from the reality of the data generating process, e.g. no additional drivers on the policy.

The `Drop` and `Binarize` columns of Table 1 show that both the covariates and the missing value structure are important in the performance of both Catboost and ResNet, respectively offering approximately a 27% and 18% improvement over a `Mean prediction` benchmark. Interestingly, neither ResNet nor Catboost is practically more accurate than the other when trained on either covariates (`Drop`) or missing value structure (`Binarize`) alone. This similarity of performance on `Drop` suggests studies comparing GBDTs and DL, mentioned in Section 1, do not have conclusions that are necessarily transferable to the context of claim reserve modelling. Those studies found a performance edge for GBDTs, notably either ignoring or in the absence of missing data. Whereas we find, for this micro-level reserving dataset, Catboost and ResNet are practically the same in terms of accuracy with missing data ignored, due to `Drop`.

However, the `LT Min Impute` and `Mean Impute` columns of Table 1 shows that Catboost with `LT Min Impute` both achieves the best overall performance and has a better accuracy than both Our ResNet and the RTDL

ResNet under either imputation scheme. We note that this better accuracy is between well-tuned ResNets and a default Catboost, underscoring the robustness of Catboost. The improved accuracy may be because Catboost better leverages both covariate signal and missing value structure, at least under a `LT Min Impute` or `Mean Impute` strategy. The strongest performance being achieved under `LT Min Impute` could also be due to the fact that minimum imputation intuitively lends itself to the partitioning strategies of a GBDT algorithm. It is unclear how minimum imputation would interact with ResNets, nor DL more broadly. This discrepancy in performance suggests an indicator of when Catboost may outperform ResNet for micro-level reserving: when there is a high proportion of missing values. This indicator is particularly relevant as missing values are pervasive in real world datasets, and are currently not studied in their role in micro-level reserving nor their contribution to GBDT and DL performance.

These results do not rule out the potential for ResNets to perform comparably or even better than Catboost given a suitable imputation strategy. However, there does not appear to be a scientific consensus on simple and robust neural network appropriate imputation strategies. Some generative imputation approaches exist (Yoon et al., 2018) but these involve considerable extra complication and have not shown much industry adoption.

## 6 Conclusion

In this paper we investigated four different imputation schemes, described in Section 4.2.1, for handling missing values using gradient boosted decision tree and deep learning models. We investigated the impact of these imputation schemes using a large, real world insurance dataset with *not* MCAR missing values. Under two of the missing value handling schemes there is no noticeable difference between Catboost and the ResNets. However, using imputation, Catboost substantially outperforms the other models. This result highlights the importance of missing value handling in claim reserving.

More broadly, this result adds a case study to the body of evidence that gradient boosted decision trees can outperform deep learning for tabular data, but emphasises the importance of missing values in drawing this conclusion. The significant impact of missing value handling on accuracy also suggests that when analysing datasets with missing values, extra care should be taken choosing the missing value handling method and not to just focus on model selection. Furthermore, our results suggest research comparing gradient boosted decision trees to deep learning for tabular data could benefit from including more datasets with missing values, especially missing values that are *not* MCAR. The importance of including missing value handling in comparisons was corroborated on other datasets using the Tabzilla repository (McElfresh et al., 2024) by injecting MCAR missing values, described in Appendix A.6. However, with MCAR missing values the effect of missing value handling was lessened.

Based on the analysis in this work we recommend exercising caution when using deep learning models for claim reserving as they require thorough tuning and their interaction with imputation schemes is not understood. Furthermore, using Catboost for claim reserving has some practical advantages, beyond potentially better accuracy with the correct missing value handling. Catboost is fast, robust and easy to use off-the-shelf. In comparison, deep learning methods require more expertise to deploy successfully: requiring both architecture selection and hyperparameter tuning. Even with the expertise required to select, implement and optimise deep learning models there is no compelling empirical or theoretical evidence that they are likely to produce better results for claim reserving.

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

# A Appendices

## A.1 Data properties

This appendix presents further properties of the data.

### A.1.1 Missing value concentration

Tables 2 and 3 characterise the concentration of missing values in our dataset.

| Fraction $f$ missing | Percentage of columns with given fraction missing |
| --- | --- |
| $f = 0$ | 45 |
| $0 < f \leq 0.2$ | 17 |
| $0.2 < f \leq 0.4$ | 13 |
| $0.4 < f \leq 0.6$ | 11 |
| $0.6 < f \leq 0.8$ | 7 |
| $0.8 < f \leq 1.0$ | 6 |

Table 2: Missingness distribution by column, to nearest %

| Fraction $f$ missing | Percentage of rows with given fraction missing |
| --- | --- |
| $0 < f \leq 0.2$ | 46 |
| $0.2 < f \leq 0.4$ | 19 |
| $0.4 < f \leq 0.6$ | 12 |
| $0.6 < f \leq 0.8$ | 11 |
| $0.8 < f \leq 1.0$ | 11 |

Table 3: Missingness distribution by row, to nearest %

### A.1.2 Time-varying properties

Figure 4 depicts the mean claim value per month, with claim value and year of incident anonymised for commercial confidentiality. The figure shows that mean claim value is clearly dependent on the time of the claim – exhibiting both seasonality and trend. This has implications for both the evaluation of model accuracy and encoding of time data.

When evaluating the accuracy of the ML models, described in Section 3, the time series nature of the data requires the definition of the test set data to be in the future relative to all training and validation data. This is to prevent bias in the estimation of performance, a form of data leakage (Nisbet et al., 2009, p. 742).

When considering how to encode the time variables, we aim to encode in such a way as to make it easier to fit seasonality and trend. To explicitly encode cyclic timestamp properties, we encode month and day-of-month variables separately. Having cyclic values in the input data, like month and day-of-month, intuitively makes it easier to fit conditional on cyclic seasons. In order to fit overall trend, we chose to also encode timestamps as a monotonic value, enabling the model to order claims in time from a single numeric value. We did this at two resolutions: i) year and ii) seconds since the epoch, also known as *Unix time*. The goal of encoding

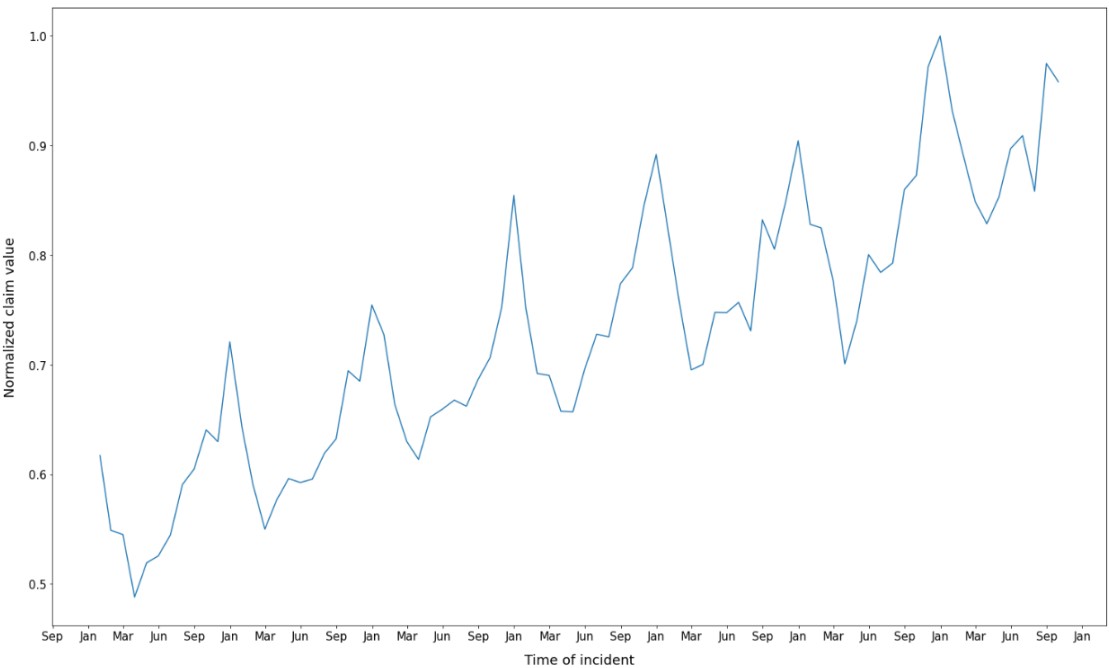

Figure 4: Time series plot of mean standardised claim values against anonymised time. Claim values were aggregated on a month. Claim values and years of data are anonymised for confidentiality.

in years is to allow a fit to large scale trends such as inflation. The goal of encoding in seconds is to enable more granular ordering in time. Therefore, overall we encode a single timestamp feature into 4 variables: year, month, date and seconds since the epoch.

### A.1.3 High-cardinality categorical variables

The dataset contains some high-cardinality categorical variables: insuree car make with hundreds of categories, insuree car model with thousands of categories and *first half* of insuree postcode with thousands of categories. The typical one-hot encoding (Murphy, 2022, p. 23) of these features would dramatically blow up the dataset size in number of features. Therefore, we applied numeric encoding strategies such as impact encoding (Pargent et al., 2022), also known as *target mean encoding*, and Catboost's novel categorical encoding, described in Section 3.2.2.

As the UK has 1.7 million postcodes (ONS, 2024), attempting to treat full postcodes as a categorical variable would give rise to at least hundreds of thousands of categories. Even target mean encoding such a high cardinality categorical could exhibit high bias and high variance due to the low number of samples per category; following the intuition outlined in Prokhorenkova et al. (2018). Only modelling the first half of postcodes was used to decrease the cardinality of the variable to thousands of categories to potentially reduces the bias and variance of the mean encoding.

### A.2 Training speed

Using 12 Intel Xeon 6136 CPUs and approximately 60GB of RAM: Catboost trains on our dataset in the order of 3 minutes per training run and the ResNet MLP architecture trains in the order of 45-120 minutes per training run. TabNet trains in the order of 300 minutes per training run. Although both Catboost and ResNet exhibited training times low enough to be retrained many times a day to keep up with changes in underlying distribution; the significantly faster training time of Catboost enables greater experimentation. This is potentially relevant for other applications as it has been shown that the difference

in modelling strategies without hyperparameter tuning is just as great as the difference within a model class with hyperparameter tuning when training on tabular data (Kadra et al., 2021).

### A.3 Differences with McElfresh

There are a few key differences between Our ResNet MLP and the RTDL ResNet MLP:

- Our ResNet MLP HPO tunes more HPs; McElfresh et al. (2024) only tunes learning rate whereas our ResNet uses an HPO grid containing learning rate, weight decay, regularisation coefficient, learning rate schedule and optimizer.

- Our ResNet MLP does not attempt to embed categorical features, as target mean encoding was used to convert all categories into scalar representations.

- Our ResNet MLP has skip/residual connections from output of each block to the overall model output; whereas residual connection only skip blocks in McElfresh et al. (2024).

- We use three repeating blocks prior to the regression head layer; McElfresh et al. (2024) use two.

- Our ResNet MLP does not vary dimensionality of the hidden state - we keep constant dimensionality of 256, in comparison to McElfresh et al. (2024) who step down to 128 and back up to 256 throughout the network. Ultimately, this and the above point result in our model having approximately 2.25x the parameters of McElfresh et al. (2024).

### A.4 Optuna HPO Results

Table 4 shows a table analogous to Table 1 comparing results obtained using Optuna for the HPO instead of grid search. It can be seen that there is no substantive difference to the conclusions in Table 1 of the work from swapping to Optuna HPO; as core points are made in comparison of Our ResNet MLP and Catboost (which did not undergo HPO).

Table 4: Results when using Optuna HPO

| Model | Drop | $(\pm se)$ | LT Min Impute | $(\pm se)$ | Mean Impute | $(\pm se)$ | Binarize | $(\pm se)$ |
|---|---|---|---|---|---|---|---|---|
| Our ResNet MLP + Grid Search (ME) | 2046 | $(\pm 1.2)$ | 1872 | $(\pm 4.5)$ | 7855 | $(\pm 1600)$ | 2288 | $(\pm 1.8)$ |
| Our ResNet MLP + Optuna HPO (ME) | 2039 | $(\pm 5.1)$ | 1887 | $(\pm 6.5)$ | 7673 | $(\pm 1600)$ | 2283 | $(\pm 1.3)$ |

### A.5 Impact of imputation scheme on HPO

This appendix presents preliminary experimental data used to justify the process of performing HPO independently of imputation scheme; as described in the beginning of Section 4. The closeness in parameter magnitude value and the robustness of HPO performance to varying hyperparameters suggested it was acceptable to reduce the computational complexity of experiments by performing HPO under one imputation scheme and evaluating on all. The LT Min Impute scheme was chosen as initial results suggested it would be the scheme with the best performance; and as such chosen in an attempt to give a level playing field for best performance across models.

#### A.5.1 Grid search based

The optimum hyperparameters obtained using grid search with LT Min Impute and Drop for our ResNet from preliminary analysis are presented in Table 5. Due to similarity in the optimal hyperparameters obtained it was concluded from this preliminary analysis that HPO could be performed independent of imputation scheme; keeping overall computational cost down.

Table 5: Best performing hyperparameters under different imputation schemes; following a grid search HPO procedure.

| Hyperparameter | LT Min Impute | Drop |
|---|---|---|
| Regularisation coefficient | 0.01 | 0.01 |
| Initial learning rate | 0.01 | 0.01 |
| Learning rate schedule | CosineDecay | CosineDecay |
| Optimiser | RMSProp | RMSProp |
| Weight Decay | 0.01 | 0.0001 |

### A.5.2  Optuna based

The optimum hyperparameters obtained using Optuna with `LT Min Impute` and `Drop` for our ResNet are presented in Table 6. With corresponding final outcomes presented in Table 7. It can be seen from 7 that although tuning with `Drop` gives different hyperparameters; and gives our ResNet more stable results for `Mean Impute`; the relative rankings in the final Table 1 would be unaffected.

Table 6: Best performing hyperparameters under different imputation schemes; following an Optuna HPO procedure.

| Hyperparameter | LT Min Impute | Drop |
|---|---|---|
| Regularisation coefficient | 0.017 | 0.68 |
| Initial learning rate | 0.009 | 0.006 |
| Learning rate schedule | CosineDecay | ExponentialDecay |
| Optimiser | RMSProp | RMSProp |
| Weight Decay | 0.0002 | 0.0002 |

Table 7: Results when using Optuna HPO

| Model | Drop | $(\pm se)$ | LT Min Impute | $(\pm se)$ | Mean Impute | $(\pm se)$ | Binarize | $(\pm se)$ |
|---|---|---|---|---|---|---|---|---|
| Our ResNet MLP + LT Min Impute Optuna HPO | 2039 | $(\pm 5.1)$ | 1887 | $(\pm 6.5)$ | 7673 | $(\pm 1600)$ | 2283 | $(\pm 1.3)$ |
| Our ResNet MLP + Drop Optuna HPO | 2047 | $(\pm 4.3)$ | 1885 | $(\pm 4.6)$ | 4548 | $(\pm 680)$ | 2292 | $(\pm 2.0)$ |

### A.6  TabZilla replication

This appendix details the results obtained from comparing `LT Min Impute`, `Mean Impute` and `Drop` on some extra datasets and algorithms from the TabZilla Benchmark (McElfresh et al., 2024). Source code can be found at `https://github.com/paper3193/tabzilla`.

30% of numeric values were randomly replaced with missing values from each dataset and then imputed using the imputation schemes studied. Then the corresponding OpenML task was performed using Catboost, XGBoost, FTTransformer (Gorishniy et al., 2021) and the RTDL ResNet. We present the results by dataset in Table 8.

From Table 8 we can see that the overall impact of imputation schemes is not large and in some cases causes no performance difference between certain imputation schemes. However, the changes in accuracy rankings of the datasets under different imputation schemes demonstrates that in principle it is possible for the rankings, and therefore results such as those in McElfresh et al. (2024), to be influenced by imputation scheme. We note that, by design, Table 8 shows results from injected MCAR missing values and as such we

do not attempt to interpret the absolute performance of any imputation scheme as MCAR data is not the focus of our work.

In summary, the TabZilla replication with different imputation schemes demonstrates in principle that handling missing values could be important; but the MCAR nature of the injected missing values and low signal value of the numerics in the chosen datasets produces a less pronounced effect than our work with non MCAR claim reserving. This highlights the importance of analyses using real world large datasets with not MCAR missing value structure in comparing GBDTs and DL models.

Table 8: Relative accuracy and ranking per dataset per algorithm using the TabZilla repository under different imputation schemes. We report TabZilla mean 10-fold cross-validation test accuracy; as extracted from the tuned aggregated results output.

| Dataset | Model | Mean Impute | | Drop | | LT Min Impute | |
|---|---|---|---|---|---|---|---|
| | | Accuracy | Rank | Accuracy | Rank | Accuracy | Rank |
| ada_agnostic | CatBoost | 0.857 | 1 | 0.854 | 2 | 0.857 | 1 |
| ada_agnostic | FTTransformer | 0.844 | 4 | 0.845 | 3 | 0.846 | 3 |
| ada_agnostic | RTDL ResNet | 0.846 | 3 | 0.842 | 4 | 0.841 | 4 |
| ada_agnostic | XGBoost | 0.855 | 2 | 0.855 | 1 | 0.855 | 2 |
| LED-display | CatBoost | 0.716 | 2 | 0.714 | 3 | 0.716 | 3 |
| LED-display | FTTransformer | 0.708 | 4 | 0.728 | 1 | 0.728 | 1 |
| LED-display | RTDL ResNet | 0.724 | 1 | 0.724 | 2 | 0.722 | 2 |
| LED-display | XGBoost | 0.710 | 3 | 0.706 | 4 | 0.710 | 4 |

## A.7 Combining missing value handling strategies

Table 9 shows the performance obtained when combining both the LT Min Impute and Binarize approaches, turning each $D$ dimensional feature vector into a $2D$ dimensional feature vector. For each original feature we generate two features: one following LT Min Impute and another that is a binary indicator of whether the data is (Binarize). This could be interpreted as providing both features and 'missingness masks' of present features to the model. Table 9 also includes results for LT Min Impute and Binarize from Table 1 for reference. Although it is a natural question to investigate combinations of compatible methods, we find that for our dataset there is no noticeable difference between combining LT Min Impute and Binarize and using LT Min Impute on its own.

Table 9: $\mathcal{D}^{\text{test}}$ RMSE mean and standard error, $se$, over cross validation partitions. **Lower RMSE is better**. RMSE is to the nearest integer, $se$ is to 2 significant figures. (CE) indicates Catboost encoding was used for categorical encoding. ‡ corresponds to an Optuna HPO scheme.

| Model | LT Min Impute | $(\pm se)$ | Binarize | $(\pm se)$ | LT Min Impute + Binarize | $(\pm se)$ |
|---|---|---|---|---|---|---|
| Catboost (CE) | 1452 | $(\pm 6.3)$ | 2302 | $(\pm 0.18)$ | 1449 | $(\pm 1.6)$ |
| Catboost | 1574 | $(\pm 2.1)$ | 2268 | $(\pm 0.28)$ | 1575 | $(\pm 2.8)$ |
| Our ResNet | 1872 | $(\pm 4.5)$ | 2288 | $(\pm 1.8)$ | 1883 | $(\pm 8.9)$ |
| RTDL ResNet‡ | 2007 | $(\pm 20)$ | 2282 | $(\pm 2.4)$ | 1994 | $(\pm 18.7)$ |
| Mean prediction | 2764 | $(\pm 0.020)$ | 2764 | $(\pm 0.020)$ | 2764 | $(\pm 0.020)$ |

