# OpenReview forum: "Investigating the impact of missing value handling on Boosted trees and Deep learning for Tabular data: A Claim Reserving case study"
_TMLR — Accepted by TMLR_

### Review · Reviewer_PdDC · 2024-09-24

**Summary Of Contributions:**

The authors study the problem of claim reserving in car insurance, i.e estimating the cost of insurance claims already reported but not yet settled. In particular, the authors study the performance of neural networks (here a Resnet and TabNet, compared to Gradient Boosting Decision trees (here, CatBoost) with light hyperparameter tuning, on a new (not used in previous publications as far as I understand) commercial dataset of car insurance.

The paper report very poor performance of TabNet on this dataset, and quite similar performance for ResNet and CatBoost when dropping missing value, or predicting using only the missing data structure. However, using a different missing value encoding ("less-than-minimum" encoding) makes CatBoost much better than the ResNet model (though it helps both models), highlighting the importance of missing value imputation strategies when comparing different models.

**Audience:**

Yes

**Broader Impact Concerns:**

No concerns.

**Claims And Evidence:**

Yes

**Requested Changes:**

# Critical requested changes

Try different imputation strategies, in particular mean imputation and missingness masks.

Show the performance of a standard pipeline on this dataset for reference. For instance, take the hp spaces from McElfresh et al. for CatBoost and ResNet and do light hpo.

Details validation strategy with respect to time. Currently it is not clear if the validation sets are also taken in the future.

# Non-critical change

Evaluate more models.

Maybe only drop rows in the "drop" strategy.

The current explanation of the "binarize" strategy is a bit hard to understand.

> Finally, Section ?? touches on training speeds
as a practical consideration.

broken reference

**Strengths And Weaknesses:**

# Strength

The authors use a new (large) dataset, which has not been used in previous publications (if I understand correctly). As such, it is useful to check the results of previous comparisons, and provide additional information.

Studying the impact of imputation strategy on different models (neural networks, GBDTS..) is underexplored in current comparisons and an interesting avenue of research.

The paper is well written. In particular, the descriptions of the model provide good short summaries.

# Weaknesses

Using only one dataset makes it hard to generalize the conclusion of the paper, in particular the discussion on imputation strategy.

The number of models compared is quite limited. In particular, TabNet is known to perform very poorly in general (as found and noted in the current paper), and the previous paper on insurance severity finding good performance for TabNet is very poorly done. This is not an issue, but it means that only 2 interesting models are compared. Other GBDTS (XGBoost, LightGBM...), and other neural networks architecture (TabR[1], FT-Transformer[2], numerical embeddings[3], RealMLP[4]...) could be investigated. That being said, we find ResNet to be a good choice if a single neural networks is chosen, as, as noted in the current paper, it was the best performing neural network in McElfresh et al.

Continuing on the last point, the issue is that the authors do not follow the guidelines of McElfresh et al. (which probably contribute to the good performance of ResNet in their paper, as ResNet do not performance so well in other comparisons, e.g Grinsztajn et al.) for their ResNet model. In particular, they do not use the same hyperparameter space nor HPO strategy, and do not use the same imputation strategy. I am not sure whether the model implementation is taken from McElfresh et al. I understand that it's likely possible to do better than McElfresh et al. when focusing on 1 dataset, but it would be interesting to compare to the standard strategy of McElfresh et al., to check that the authors' results are indeed better.

The imputation strategy comparison is quite unfair to the neural networks, as the only imputation strategy tried is "less-than-minimum" imputation, CatBoost default strategy. As such, it is perhaps less surprising that CatBoost benefits the most from this strategy. Other imputation strategies could be easier to use for neural networks, such as mean imputation, used in McElfresh et al. Adding missingness masks could also be interesting, and perhaps more useful for neural networks.

The HPO strategy is perhaps suboptimal, as grid search is often a bad choice compared to random search on a larger set of hyperparameters. Again, I can imagine that the authors have used more manual tuning to restrict the hp space on their one dataset, and found good hp, but it is not clear if the manual effort is fair when comparing different models. For this reason, it would be interesting to compare the performance with more standard hp space and random search or other hpo algorithm.

I wonder if the results would be easier to interpret for "drop" if only rows were removed, and sample sizes were equalized (it won't be comparable to other strategies in any cases, but it might be easier to compartmentalise the performance in "drop" + "binarize").

[1]: Gorishniy, Yury, Ivan Rubachev, Nikolay Kartashev, Daniil Shlenskii, Akim Kotelnikov, and Artem Babenko. 2023. “TabR: Tabular Deep Learning Meets Nearest Neighbors in 2023.” arXiv:2307.14338.

[2]: Gorishniy, Yury, Ivan Rubachev, Valentin Khrulkov, and Artem Babenko. 2023. “Revisiting Deep Learning Models for Tabular Data.” arXiv. https://doi.org/10.48550/arXiv.2106.11959.

[3]: Gorishniy, Yury, Ivan Rubachev, and Artem Babenko. 2023. “On Embeddings for Numerical Features in Tabular Deep Learning.” arXiv. https://doi.org/10.48550/arXiv.2203.05556.

[4]: David Holzmüller, Léo Grinsztajn, Ingo Steinwart. 2024. "Better by Default: Strong Pre-Tuned MLPs and Boosted Trees on Tabular Data." arXiv. https://arxiv.org/abs/2407.04491

---

> ### Author Response · Authors · 2024-10-28
> **Response to reviewer PdDC**
>
> We thank the reviewer for their detailed feedback and are grateful for the constructive suggestions to improve our work.
> We are happy to see the reviewer appreciates the writing and relevance of our work.
>
> We respond to the reviewer's comments below.
>
> # [Point of Clarity]
> The reviewer notes:
> > (not used in previous publications as far as I understand) commercial dataset of car insurance.
>
> Indeed this dataset has never been analysed in any other publication. We have added clarification to Section 2.1.
>
> # [Weakness 1]
> > Using only one dataset makes it hard to generalize the conclusion of the paper, in particular the discussion on imputation strategy.
>
> Please see comments under "1. More datasets" in our general comment.
>
> # [Weakness 2]
> > The number of models compared is quite limited.
> > ...
> > it means that only 2 interesting models are compared. Other GBDTS (XGBoost, LightGBM...), and other neural networks architecture (TabR[1], FT-Transformer[2], numerical embeddings[3], RealMLP[4]...) could be investigated. That being said, we find ResNet to be a good choice if a single neural networks is chosen, as, as noted in the current paper, it was the best performing neural network in McElfresh et al
>
> Please see comments under "3. More model baselines" in our general comment.
>
> We would like to note that while we agree there are other architectures and models that may perform better in accuracy; our goal with this work is to focus on the importance of imputation in real applications.
> Therefore we are happy the reviewer agrees with our model selection choices.
>
> # [Weakness 3]
> >Continuing on the last point, the issue is that the authors do not follow the guidelines of McElfresh et al. (which probably contribute to the good performance of ResNet in their paper, as ResNet do not performance so well in other comparisons, e.g Grinsztajn et al.) for their ResNet model. In particular, they do not use the same hyperparameter space nor HPO strategy, and do not use the same imputation strategy. I am not sure whether the model implementation is taken from McElfresh et al. I understand that it's likely possible to do better than McElfresh et al. when focusing on 1 dataset, but it would be interesting to compare to the standard strategy of McElfresh et al., to check that the authors' results are indeed better.
>
> Please see comments under "4. More standard HPO" in our general comment.
> We agree that contextualising this work closer to McElfresh et al. [1] would strengthen our contribution, so we have altered Section 3.3 to clarify that our ResNet architecture was not identical to that of Gorishniy et al [2] as used in McElfresh et al. [1].
> Furthermore, we extend our analysis with the architecture of Gorishniy et al [2] to allow interpretation of our work in the broader literature, updating Table 1.
>
> # [Weakness 4]
> >The imputation strategy comparison is quite unfair to the neural networks, as the only imputation strategy tried is "less-than-minimum" imputation, CatBoost default strategy. As such, it is perhaps less surprising that CatBoost benefits the most from this strategy. Other imputation strategies could be easier to use for neural networks, such as mean imputation, used in McElfresh et al. Adding missingness masks could also be interesting, and perhaps more useful for neural networks.
>
> Please see comments under "2. More imputation schemes" in our general comment.
> We have added results from using mean imputation.
>
> # [Weakness 5]
> > The HPO strategy is perhaps suboptimal, as grid search is often a bad choice compared to random search on a larger set of hyperparameters. Again, I can imagine that the authors have used more manual tuning to restrict the hp space on their one dataset, and found good hp, but it is not clear if the manual effort is fair when comparing different models. For this reason, it would be interesting to compare the performance with more standard hp space and random search or other hpo algorithm.
>
> Please see comments under "4. More standard HPO" in our general comment.

---

> ### Author Response · Authors · 2024-10-28
> **Continuation of response to reviewer PdDC**
>
> # [Weakness 6]
> > I wonder if the results would be easier to interpret for "drop" if only rows were removed, and sample sizes were equalized (it won't be comparable to other strategies in any cases, but it might be easier to compartmentalise the performance in "drop" + "binarize").
>
> We are grateful for the reviewer's clear engagement with our work by taking the time to give interesting suggestions.
> There appears to be two aspects to the reviewer's suggestion:
>
> 1. Drop 'only rows'. By this we assume the reviewer meant to avoid dropping columns prior to dropping rows. We would point out that _every_ row has at least one missing value due to the overlapping nature of the missing values. We have attempted to clarify this with our edit in Section 2.1.1.
>
> 2. Equalize dataset sample sizes between `Drop` and `Binarize`. Although this may allow for some more 'apples-to-apples' comparison between `Binarize` and `Drop` it is unclear how this would better compartmentalise the effects of fitting to covariates and fitting to missingness structure. We appreciate that the performance of `Binarize` could be partly driven by inclusion of more data, but that is exactly why we would want to include missing values in our analysis - allowing us to see the improvement gain from just missing values over present covariates.
>
> # [Requested Change 1]
> > Try different imputation strategies, in particular mean imputation and missingness masks.
>
> As above; we are grateful for this suggestion and added results with mean imputation.
>
> # [Requested Change 2]
> > Show the performance of a standard pipeline on this dataset for reference. For instance, take the hp spaces from McElfresh et al. for CatBoost and ResNet and do light hpo.
>
> We addressed this point in our answer to *[Weakness 3]* and *[Weakness 4]*.
> To summarise:
> We have rerun our ResNet model using Optuna for HPO with the 30 trials as in McElfresh et al. [1] - as the results are not substantively different they are presented in Appendix A.4.
> We have also implemented the ResNet architecture of Gorishniy et al. [2] and perform HPO with an identical grid to that of McElfresh et al. [1] to present results in Table 1.
>
> # [Requested Change 3]
> > Details validation strategy with respect to time. Currently it is not clear if the validation sets are also taken in the future.
>
> The validation sets were randomly sampled in time, excluding the timeframe set aside for test sets.
> We have added clarification to the manuscript in Section 4.1 and in the caption of a redesigned Figure 2.
>
> # [Non-critical changes]
> > Evaluate more models.
>
> We address points on more datasets in our general comment.
>
> > Maybe only drop rows in the "drop" strategy.
>
> We have addressed this point in our answer to *[Weakness 6]*.
>
> > The current explanation of the "binarize" strategy is a bit hard to understand.
>
> We thank the reviewer for helping us strive for clarity.
> We have added Figure 3; we hope this addresses the reviewer's point.
>
> > broken reference
>
> We fix the error in the rendering of our manuscript.

---

### Review · Reviewer_rES8 · 2024-09-30

**Summary Of Contributions:**

This paper explores the ongoing debate between deep learning (DL) models and gradient boosted decision trees (GBDTs) when it comes to handling tabular data. Focusing on a claim reserving application from a large UK insurer, the authors compare three models: TabNet, a specialized DL architecture; a simple neural network inspired by ResNet; and Catboost, a popular GBDT model. They aim to illuminate how missing value handling can significantly impact model accuracy—a topic often overlooked in previous research.

The findings reveal that with some imputation methods, the optimized simple neural network performed comparably to Catboost. However, other imputation methods revealed Catboost outperformed both DL models, achieving the best accuracy overall. This underscores the critical role that missing value handling plays in model performance. Ultimately, the study suggests that before jumping to conclusions about which algorithm reigns supreme for tabular data, we must first address how missing values are managed.

**Audience:**

Yes

**Broader Impact Concerns:**

None.

**Claims And Evidence:**

No

**Requested Changes:**

Suggested edits which would greatly impact my evaluation of the paper:
- Can the authors consider expanding this work to other datasets as well? I understand the benefit of the data that you currently use may be that it is commercially viable (used by a company in real world setting), but one thing you could do is to replicate the TabZilla project (or part of it) with removing some data.
- Can the authors incorporate some more standard HPO (and ideally) NAS+HPO methods (like from the [optuna](https://optuna.org)) framework? Using more rigorous and widely-accepted methods of hyperparameter searches would improve the results of the paper significantly.

Suggested edits which would improve the view of the paper:
- As noted in the weaknesses above, the authors could improve the writing and description of some of the conclusions of the work. In addition to those previously listed, the authors should consider making the tables and figures more helpful for the reader. For example, Figure 2 takes up an entire page, but it does not provide much description besides what is already described in the work and I do not feel advances the reader's understanding of the work. A figure about the hyperparemetters and tuning procedure however would be much more helpful. Additionally, the paper's structure currently reads with the first 6 pages being mostly background and prep material. I'd suggest compressing those sections and building out the last 3 sections with more details and experiments (as suggested above).

Very Minor Note:
- "The process of determining price for a prospective customer" -> "The process of determining the/a price for a prospective customer"
- Missing ref: "Finally, Section ?? touches on training speeds as a practical consideration."

**Strengths And Weaknesses:**

Strengths:
- The authors propose and undertake an interesting and important topic -- missing data values in tabular data. Specifically the angle of the research -- how do missing values impact the GBDT vs deep learning debate -- is important and worthy of study
- This work is clearly angled towards a specific domain which the authors have deep technical expertise in (or so it seems from reading the work, particularly Section 2). The authors communicate the various aspects of the car insurance marketplace and how the technical interventions are deployed within the larger car insurance pipeline. This is a very nice aspect of this paper.
- The paper is generally well-wrtitten and generally describes the work the authors did in an appropriate level of depth with nice clarity.

Weaknesses:
- The main weakness is the limited application of this work just to car insurance. Other work that looks at the GBDT vs DL debate in tabular data, like McElfresh et al., does so across a wide range of datasets and topic areas. This is a major weakness for the generalizability of the findings the authors gave. Particularly in light of the main conclusion from other work (like McElfresh) that the superiority of a given model greatly depends on the dataset.
- The dataset the authors use is not public. Too little information is given about the dataset. For example, we don't even know how many rows there are, what the variables are, temporal or geographic components, how concentrated the missing values are per column, etc.
- Could the authors provide more evidence to this claim: "Preliminary analysis suggested the best hyperparameters did not vary with preprocessing scheme." Like what experiments were run and what the results were?
- Grid search is the most basic and least effective algorithm for hyperparameters tuning. The authors should consider using optuna to perform at least a random search through their search space. They could perform Bayesian methods as well.
- "Although there is research on generating architectures following an algorithm these approaches are computationally expensive and give only slight performance improvements (Ren et al., 2021). Since these methods are seldom used in practice, the depth and width of our ResNet MLP was selected heuristically." I do not believe the literature supports this claim. Neural Architecture Search has been very productive in a wide range of applications [1,2,3,4], including tabular data. Can the authors provide more details for why they didn't perform any NAS or NAS+HPO experiments?

[1] https://openaccess.thecvf.com/content_ICCV_2019/html/Howard_Searching_for_MobileNetV3_ICCV_2019_paper.html

[2] https://proceedings.neurips.cc/paper_files/paper/2023/file/eb3c42ddfa16d8421fdba13528107cc1-Paper-Conference.pdf

[3] https://arxiv.org/abs/2006.02049

[4] https://arxiv.org/abs/2310.12145

---

> ### Author Response · Authors · 2024-10-28
> **Response to reviewer rES8**
>
> We thank the reviewer for their helpful feedback and thorough review.
> We are pleased to read that the reviewer appreciated the quality of our writing and especially the effort to contextualise our work in the insurance application.
> We appreciate the precise summary of our work; and believe the reviewer captured the core message of the paper well.
>
> We respond to the reviewer's comments below.
>
> # [Weakness 1]
> >The main weakness is the limited application of this work just to car insurance. Other work that looks at the GBDT vs DL debate in tabular data, like McElfresh et al., does so across a wide range of datasets and topic areas. This is a major weakness for the generalizability of the findings the authors gave. Particularly in light of the main conclusion from other work (like McElfresh) that the superiority of a given model greatly depends on the dataset.
>
> Please see comments under "1. More datasets" in our general comment.
>
> # [Weakness 2]
> >The dataset the authors use is not public. Too little information is given about the dataset. For example, we don't even know how many rows there are, what the variables are, temporal or geographic components, how concentrated the missing values are per column, etc.
>
> We understand the reviewer but our hands are tied with respect to how much detail we can give about the dataset as it is commercially sensitive.
> We note that in our original manuscript we specified the number of rows as hundreds of thousands.
> Also appendices provided more dataset description: including time varying properties of the response, as well as a discussion on the handling of postcodes (geographic) in the processing of the dataset.
> We have updated the wording in Section 2.1 to convey the number of rows is more than 100k but less than 1 million; added a more detailed description of the missing value concentrations to Appendix A.1, and signposted the relevance of this appendix in Section 2.1.
>
> # [Weakness 3]
> >Could the authors provide more evidence to this claim: "Preliminary analysis suggested the best hyperparameters did not vary with preprocessing scheme." Like what experiments were run and what the results were?
>
> We have expanded this point in Appendix A.5 and signposted this in Section 4.
> For clarity: we ran our grid search HPO with both `Drop` and `LT Min Impute` for our ResNet architecture and found the optimal HPs in both cases to be very similar.
> However, we are aware that as we did grid search (instead of random HPO) that the similarity of the optimal HPs is more likely, so we also use Appendix A.5 to present exploration of how using Optuna may impact these preliminary results.
> From this we concluded that, for at least the hyper parameters we considered tuning, it was not vital to do HPO conditional on a specific imputation scheme.
>
> We did not repeat this experiment with TabNet as we had observed the very poor performance and long training times; highlighting that the model was unlikely to be as performant as published and thus less central to the message of the paper.
>
> We did not perform HPO on Catboost.
>
> # [Weakness 4]
> > Grid search is the most basic and least effective algorithm for hyperparameters tuning. The authors should consider using optuna to perform at least a random search through their search space. They could perform Bayesian methods as well.
>
> Please see comments under "4. More standard HPO" in our general comment.
>
> # [Weakness 5]
> > "Although there is research on generating architectures following an algorithm these approaches are computationally expensive and give only slight performance improvements (Ren et al., 2021). Since these methods are seldom used in practice, the depth and width of our ResNet MLP was selected heuristically." I do not believe the literature supports this claim. Neural Architecture Search has been very productive in a wide range of applications [1,2,3,4], including tabular data. Can the authors provide more details for why they didn't perform any NAS or NAS+HPO experiments?
>
> We did not intend that the tone of our original manuscript to come across dismissive of NAS.
> We acknowledge that NAS has had success in some applications.
> The authors were simply unaware of successful commercial applications of NAS.
> We have rewritten Section 4.1.3 to reflect we have no prejudice to NAS but that NAS was not feasible given the computational resources.
>
> # [Requested Change 1]
> >Can the authors consider expanding this work to other datasets as well? I understand the benefit of the data that you currently use may be that it is commercially viable (used by a company in real world setting), but one thing you could do is to replicate the TabZilla project (or part of it) with removing some data.
>
> As stated in *[Weakness 1]*: we understand the reviewer's concerns and address this under "1. More datasets" in our general comment.

---

> ### Author Response · Authors · 2024-10-28
> **Continuation of response to reviewer rES8**
>
> # [Requested Change 2]
> > Can the authors incorporate some more standard HPO (and ideally) NAS+HPO methods (like from the optuna) framework? Using more rigorous and widely-accepted methods of hyperparameter searches would improve the results of the paper significantly.
>
> As stated in *[Weakness 4]*: we understand the reviewers concerns, and address points on HPO in our general comment.
>
> # [Requested Change 3]
> > As noted in the weaknesses above, the authors could improve the writing and description of some of the conclusions of the work.
>
> We appreciate the reviewer's attention to presentation and clarity in the work.
> We have added some changes aimed at improving the clarity throughout the manuscript.
>
> > In addition to those previously listed, the authors should consider making the tables and figures more helpful for the reader. For example, Figure 2 takes up an entire page, but it does not provide much description besides what is already described in the work and I do not feel advances the reader's understanding of the work. A figure about the hyperparemetters and tuning procedure however would be much more helpful.
>
> We have attempted to reduce the impact of Figure 2 through re-design.
> We understand that the description in the text covers the same content but re-iterating through figures can be helpful for some readers.
>
> > Additionally, the paper's structure currently reads with the first 6 pages being mostly background and prep material. I'd suggest compressing those sections and building out the last 3 sections with more details and experiments (as suggested above).
>
> We believe background material helps the reader contextualise the real application of the paper and makes it self-contained; improving the strength and relevance of our manuscript.
>
> # [Minor notes]
> Thank you for spotting the grammatical error, it has been fixed. We also fix the rendering error of our manuscript.

---

### Review · Reviewer_STkm · 2024-10-15

**Summary Of Contributions:**

The paper address the issue of missing value in the tabular datasets for deep tabular learning models.

**Audience:**

Yes

**Claims And Evidence:**

No

**Requested Changes:**

Apart from the weaknesses noted above, I would recommend the authors:

- Remove definitions of well-known algorithms: Exclude explanations for commonly understood algorithms like gradient boosting and ResNet. Similarly, consider removing Figures 1 and 2.

- Consider a more suitable venue. Given that the paper exclusively examines an insurance dataset, it may be more appropriate for a journal or conference specializing in that field.

**Strengths And Weaknesses:**

Strengths:

- The paper addresses an important issue in the field.

Weaknesses:

- Limited data analysis. Only a single dataset was used, which limits the generalizability of the findings.
- Writing clarity. The manuscript’s structure and language are often unclear, impacting readability and understanding.
- Insufficient baseline comparisons. The study evaluates only one ML model and two DL models, which is inadequate for a comprehensive analysis.
- Limited baselines for missing value handling (Section 4.2.1). Only three methods are considered, while more techniques are commonly used in practice.

---

> ### Author Response · Authors · 2024-10-28
> **Response to reviewer STkm**
>
> We thank the reviewer for their review, and for the appreciation of the relevance of our work to the field.
>
> We respond to the reviewer's comments in the points below.
>
> # [Weakness 1]
> > Limited data analysis. Only a single dataset was used, which limits the generalizability of the findings.
>
> Please see comments under "1. More datasets" in our general comment.
>
> # [Weakness 2]
> > Writing clarity. The manuscript’s structure and language are often unclear, impacting readability and understanding.
>
> We regret that the reviewer found our manuscript lacking clarity.
> The manuscript has undergone substantial revision.
>
> # [Weakness 3]
> > Insufficient baseline comparisons. The study evaluates only one ML model and two DL models, which is inadequate for a comprehensive analysis.
>
> Please see comments under "3. More model baselines" in our general comment.
>
> # [Weakness 4]
> > Limited baselines for missing value handling (Section 4.2.1). Only three methods are considered, while more techniques are commonly used in practice.
>
> Please see comments under "2. More imputation schemes" in our general comment.
>
> # [Requested Change 1]
> > Remove definitions of well-known algorithms: Exclude explanations for commonly understood algorithms like gradient boosting and ResNet. Similarly, consider removing Figures 1 and 2.
>
> We believe it is important to lay clear background to improve the self-contained nature of the work.
> For example, Figure 1 is important to add clarity on our implementation; especially as we regretfully cannot provide code due to commercial restrictions. We have redesigned Figure 2 to take less space in the manuscript.
>
> # [Requested Change 2]
> > Consider a more suitable venue. Given that the paper exclusively examines an insurance dataset, it may be more appropriate for a journal or conference specializing in that field.
>
> The authors believe this work falls precisely in the remit of TMLR; as stated in the [Submission Guidelines](https://jmlr.org/tmlr/editorial-policies.html).
> More precisely under the point: _"accounts of applications of existing techniques that shed light on the strengths and weaknesses of the methods"_.
> We study a claim reserving application to shed light on the importance of imputation schemes in model comparisons.

---

### Author Response · Authors · 2024-10-28
**General Comment**

We would like to thank the reviewers for their careful and detailed feedback; which helps us to improve the clarity and strength of our message.
We especially enjoyed reading that reviewers thought our work studied an important area of research (STkm/rES8/PdDC); was well written (rES8/PdDC); benefited from contextualisation in the wider insurance application (rES8) and covered background on models at the right level (PdDC).

We recognise a pattern in the comments of all reviewers that our study was focused in scope.
Reviewers all mentioned some combination of wanting more datasets, more imputation schemes and more model baselines.
We would like to stress that this work studies a large, real dataset with informative missing values; of which there is little comparable work in the literature.
This work is intended as a case study on the importance of missing values as an often overlooked step.
We do not make recommendations for imputation schemes or models outside of the specific application we describe, nor claim to develop or achieve SOTA modelling.
We believe that just demonstrating "before jumping to conclusions about which algorithm reigns supreme for tabular data, we must first address how missing values are managed" (quote from rES8) in one real application is a sufficiently important result.

# 1. More datasets
We appreciate that studying a single dataset may limit the generalisability of our findings.
Still, we wish to emphasise again that there are few, if any, large public regression datasets that contain clearly not MCAR missing structure.

To address the reviewers' comments we have added Appendix A.6 describing the results obtained when using `Mean Impute`, `Drop` or `LT Min Impute` imputation schemes with the TabZilla repository on a few datasets where we inject missing values with Optuna HPO, as suggested by rES8.
This fork may be viewed at https://github.com/paper3193/tabzilla.

For these datasets the effect of imputation on accuracy ranking is substantially lower than our own but this does not reduce the practical relevance of our conclusions.
We note that very few of the datasets used in McElfresh et al [1] have any substantial amount of missing values.
Thus, we must inject missing values.
Any experiments injecting missing values in an MCAR fashion cannot meaningfully replicate missingness structures.
Furthermore, engineering arbitrary missingness structure also does not have any guarantee of being relevant.
This highlights the value in studying real missingness patterns in a large and commercially relevant application.

# 2. More imputation schemes
We add mean imputation to Table 1, which was the imputation scheme in McElfresh et al [1].

# 3. More model baselines
As PdDC notes we aimed to choose the most important algorithms to compare.
Although, we understand that we only compare a select few models in our analyses.
This is from a practical perspective: building on top of prior research to avoid excessive computation.
However, we agree with points raised about the inconsistency of our ResNet architecture model with Gorishniy et al [2], and by extension McElfresh et al [1].
As such, we add the ResNet model used in Gorishniy et al [2] to our analyses and highlight the cell in the Table 1 that would correspond to McElfresh et al [1], better contextualising our findings.

# 4. More standard HPO
We agree with the reviewers that grid search is theoretically weaker than stochastic searches, such as those implemented by Optuna.
We rerun our analyses on our ResNet model using Optuna hyperparameter tuning.
We choose only to rerun on our ResNet model as the central message of the paper is derived from comparison of our ResNet result with that of an untuned Catboost.
As it happened, this did not give substantially different performance, and so we present these results in Appendix A.4 to avoid breaking the flow of the work.

[1] Duncan McElfresh, Sujay Khandagale, Jonathan Valverde, Vishak Prasad C, Ganesh Ramakrishnan, Micah
Goldblum, and Colin White. When do neural nets outperform boosted trees on tabular data? Advances
in Neural Information Processing Systems, 36, 2024.

[2] Yury Gorishniy, Ivan Rubachev, Valentin Khrulkov, and Artem Babenko. Revisiting deep learning models
for tabular data. Advances in Neural Information Processing Systems, 34:18932–18943, 2021

---

### Decision · Action_Editor_YQJt · 2024-11-27

**Recommendation:** Accept with minor revision

**Comment:**

Two of the reviewers lean toward acceptance while one leans toward rejection. The reviewers favoring acceptance report that the concerns described above are largely addressed. After reading the reviews, rebuttals, and part of the already revised paper myself, I agree with the two positive reviewers. That said, I have two requests that I ask the authors to at least seriously consider for the final version:
1. In the abstract and introduction, I think the authors could do a better job of justifying their focus on the single car insurance dataset, particularly because of the prevalence and not MCAR nature of its missing values as mentioned earlier. I think the justification is stronger in the conclusion (Section 6), and this messaging could be brought forward as well. More pointers could also be given to the Tabzilla experiment in the appendix to show that the problem appears less severe when missingness is MCAR.
1. Reviewer PdDC suggested "missingness masks" as an additional imputation method. I clarified with the reviewer that this means adding, for each feature with missing values, another binary-valued feature that indicates where the missing values are. As I understand it, this would be a combination of the "Binarize" method with one of the imputation methods. I think the addition of such a combination would further strengthen the work.

**Audience:**

This submission is of interest because it brings a previously neglected dimension of missing value handling to the ongoing debate between DL versus GBDT models for tabular data. Reviewer rES8 noted that the description of the application domain of insurance is nicely done, so this could be of independent interest to readers.

**Claims And Evidence:**

The main contribution of the submission is to show that the way in which missing values are handled can significantly affect comparisons between deep learning (DL) and gradient boosted decision tree (GBDT) models on tabular data. The authors use a case study on a real-world car insurance dataset with realistic missing values to establish their claims. The reviewers had several common concerns about whether the evidence was convincing and clear enough:
- Regarding **missing value handling methods**, I interpret this concern to be whether the methods investigated are sufficiently representative. During the rebuttal period, the authors added mean imputation results. This mostly addressed the reviewers' concerns (and I also think it was important), but please see my comments below for an additional method that the authors could consider.
- Regarding the **selection of models** and their **hyperparameter optimization**, I view this as a question of whether the DL vs. GBDT comparison is done properly. During the rebuttal, the authors added results using Optuna for HPO, which did not change their conclusions. They also added a second variant of ResNet to be more consistent with prior work of McElfresh et al.
- Regarding the **limitation to a single dataset**, the authors justified this by stating that the prevalence and not missing completely at random (MCAR) nature of its missing values (in short, the realism of the dataset) are important to really illustrate the problem with missing value handling. They added an experiment on a few datasets from the TabZilla repository, where missing values were artificially injected, showing that the effect of missing value handling is less pronounced. Two of the reviewers and I think that this addresses the concern, but please see my comments below for a recommendation on presentation.

---

> ### Author Response · Authors · 2024-12-13
> **Response to Decision**
>
> We are happy to see the AE's decision and would like thank the AE and all of the reviewers for their time and help in improving the quality of our manuscript.
>
> The camera ready version has been uploaded. We address the minor revisions suggested as follows:
>
> > In the abstract and introduction, I think the authors could do a better job of justifying their focus on the single car insurance dataset, particularly because of the prevalence and not MCAR nature of its missing values as mentioned earlier. I think the justification is stronger in the conclusion (Section 6), and this messaging could be brought forward as well. More pointers could also be given to the Tabzilla experiment in the appendix to show that the problem appears less severe when missingness is MCAR.
>
> We have stressed the not MCAR nature of the data in the abstract and stressed the value of focusing on the dataset in the Introduction (Section 1). We have also added that the effect is less severe in our MCAR Tabzilla experiments in the Conclusion (Section 6).
>
> > Reviewer PdDC suggested "missingness masks" as an additional imputation method. I clarified with the reviewer that this means adding, for each feature with missing values, another binary-valued feature that indicates where the missing values are. As I understand it, this would be a combination of the "Binarize" method with one of the imputation methods. I think the addition of such a combination would further strengthen the work.
>
> We have studied a combination of LT Min Impute and Binarize to look into "missingness masks". We present the findings in Appendix A.7 and reference to it from the end of Section 4.2.1. In summary, we found there to be no substantial difference with LT Min Impute for our claim reserving dataset.